# Interpolation Formulas for Asymptotically Safe Cosmology

**Sandor Nagy** *,† and **Kornel Sailer** †

Department of Theoretical Physics, Faculty of Science and Technology, University of Debrecen, P.O. Box 400, 4002 Debrecen, Hungary
* Correspondence: sandor.nagy@science.unideb.hu
† These authors contributed equally to this work.

**Abstract:** Simple interpolation formulas are proposed for the description of the renormalization group (RG) scale dependences of the gravitational couplings in the framework of the 2-parameters Einstein-Hilbert (EH) theory of gravity and applied to a simple, analytically solvable, spatially homogeneous and isotropic, spatially flat model universe. The analytical solution is found in two schemes incorporating different methods of the determination of the conversion rule $k(t)$ of the RG scale $k$ to the cosmological time $t$. In the case of the discussed model these schemes turn out to yield identical cosmological evolution. Explicit analytical formulas are found for the conversion rule $k(t)$ as well as for the characteristic time scales $t_G$ and $t_\Lambda > t_G$ corresponding to the dynamical energy scales $k_G$ and $k_\Lambda$, respectively, arising form the RG analysis of the EH theory. It is shown that there exists a model-dependent time scale $t_d$ ($t_G \leq t_d < t_\Lambda$) at which the accelerating expansion changes to the decelerating one. It is shown that the evolution runs from a well-identified cosmological fixed point to another one. As a by-product we show that the entropy of the system decreases monotonically in the interval $0 < t \leq t_\Lambda$ due to the quantum effects.

**Keywords:** renormalization group

## 1. Introduction

Asymptotically safe cosmology [1–6] relies on the success of the asymptotic safety scenario [7] in quantum gravity achieved in the last two decades (see the status reports in [8–10] and the references therein). Based on the RG studies of the four-dimensional two-parameter EH gravity in the continuum, good evidence is found for the existence of an ultraviolet (UV) fixed point, the so-called Reuter fixed point, and that of the Gaussian fixed point [11–21]. It is assumed that the early time evolution of the Universe was governed by the Reuter fixed point, revealing asymptotic safety, while its evolution preceding somewhat the present-day time was dictated by the perturbative regime of the gravitational couplings near the Gaussian fixed point. The idea is that at any cosmological time $t$, the relevant physical processes in the Universe are those of a given energy scale $k = k(t)$ identified with the RG scale $k$. The function $k(t)$ is called below the $k$-to-$t$ conversion rule. As the authors pointed out in [22], the main features of the RG flow of the gravitational couplings are rather independent of the fine details of RG scheme used, as well as the matter content of the Universe. Moreover, the RG analyses of the Euclidean and Lorentzian theories yield rather similar results in that respect [23–27]. These features of the RG flow on realistic RG trajectories include *(i)* three scaling regions, the UV regime governed by the Reuter fixed point, the crossover regime ended in the perturbative one close to the Gaussian fixed point, and the IR region where the gravitational couplings become almost scale-independent, and *(ii)* the order of magnitude of their limit points at $k_G \sim \mathcal{O}(m_{\text{Pl}})$ and $k_\Lambda \sim \mathcal{O}(10^{-30} k_G)$, where $m_{\text{Pl}}$ is the Planck mass. For the expanding Universe, one naturally expects that the decreasing energy scale $k$ should correspond to the increasing cosmological time $t$. The RG studies prove that Newton's gravitational coupling $G$ vanishes

at the Reuter fixed point, making it plausible that the quantum-improved evolution equations and the quantum-improved Friedmann equations should keep their classical form with the only modification that Newton's gravitational constant $G_0$ and the cosmological constant $\Lambda_0$ (the null index indicates the present-day values) should be replaced by the time-dependent couplings $G(t) \equiv G(k(t))$ and $\Lambda(t) \equiv \Lambda(k(t))$, respectively. From now on, we assume that the functions $G(k)$ and $\Lambda(k)$ are well-known functions from the RG analysis of quantum gravity.

So far, quantum fluctuations of the metric and those of the matter fields have been neglected, and the cosmological evolution of the homogeneous and isotropic Universe is described by the quantum modified Friedmann equations and their consistency condition, where the gravitational constants are replaced by their time-dependent counterparts. These equations represent the symmetry-reduced sector of the quantum-improved version of classical EH gravity, when the gravitational constants $G_0$ and $\Lambda_0$ are replaced by their time-dependent counterparts. Then, the Bianchi identity $\nabla_\mu G^{\mu\nu} = 0$ (for the Einstein tensor $G^{\mu\nu}$) implies the quantum-improved consistency condition $\nabla_\mu[G(t)T^{\mu\nu}] = 0$ of the Einstein equations with the stress–energy tensor of matter $T^{\mu\nu}$. Having performed the symmetry reduction to the homogeneous and isotropic sector, one obtains the quantum-improved consistency condition (cf. (13) below) of the quantum-improved Friedmann equations, which is now different from the law of the local energy conservation of matter, as it was in the classical case. Let us say that one formulates the cosmological evolution problem in terms of the Hubble parameter $H(t)$, the energy density of matter $\rho^{(m)}(t)$, and the function $k(t)$. Then, one has only two independent equations for the determination of three yet-unknown functions. Regarding this problem, various approaches have been worked out in the literature. The first attempts assumed that one has to make some intuitive assumption on the function $k(t)$ such as $k = \xi/t$ with some constant $\xi$, require the local conservation of the energy of matter separately as in classical cosmology [28], and adjust the constant $\xi$ in the UV and in the perturbative regimes separately in order to achieve consistency among the four equations [29,30]. These first efforts have given a hint about the scale- or time-dependence of $\xi$ itself: different results have been obtained for $\xi$ in the UV and in the perturbative regimes. There is an approach that determines the function $k(t)$ from the interplay between the local energy conservation of matter and the reduced consistency condition (cf. (14) below) following from the quantum-improved consistency condition [31]. Below, we shall follow this route under Scheme A. Another approach has been proposed in the framework of dimensionless cosmological variables [32,33], when the $k$-to-$t$ conversion rule is determined from a constraint on the RG parameters (cf. (24) below), which follows from taking the first derivative of the Friedmann constraint with respect to the RG scale $k$ [1]. This approach is particularly adequate to consider models in which both gravitation and matter underlie quantum effects, but we restrict now ourselves to a model where matter is represented by a barotropic fluid with the classical equation of state (EOS). Below, we shall follow this route under Scheme B. Finally, we have to mention the approach when one gives up the requirement of the local energy conservation of matter separately, then one has to go back to some intuitive choice of the conversion rule $k$-to-$t$ and establish that matter is expanding in a nonadiabatic manner, and the entropy production can be read off from the thermodynamical reinterpretation of the quantum-improved consistency condition [3]. We should mention that, in particular systems, both Scheme A and Scheme B may reproduce the naive inverse proportionality $k \propto 1/t$, but not in general. It has been shown that, according to the matter content of the Universe, the cosmological evolution may exhibit cosmological fixed points, where the RG scale freezes in, and ones where the RG scale continues to evolve; a detailed classification of the possible cosmological fixed points can be found in [1].

The main goal of the present paper was to put forward interpolation formulas that describe the above-mentioned main features of the RG flow of the gravitational couplings. Then, we shall apply these formulas to a particularly simple, analytically solvable model Universe, in order to obtain analytic relations for the characteristic time scales of the

cosmological evolution. We restricted ourselves to the asymptotically safe cosmology based on the two-parameter EH gravity. We intended to give interpolation formulas that reproduce the main features of the RG flow discussed in detail in [22]. The UV scaling for $k > k_G$ is governed by the Reuter fixed point, while for $0 < k \leq k_\Lambda$, the gravitational couplings take their present-day constant values $G_0 = m_{\mathrm{Pl}}^{-2}$ measured at the laboratory scale $k_l \approx 8.2 \times 10^{-34} m_{\mathrm{Pl}} \approx 10^{-3} k_\Lambda$ and $\Lambda_0 \approx 2.7 \times 10^{-122} m_{\mathrm{Pl}}^2$ observed at the Hubble scale $k_H \approx 8.2 \times 10^{-62} m_{\mathrm{Pl}} \ll k_\Lambda$. (It should be noticed that, along the RG trajectories relevant for the evolution of our Universe, there may exist a deep IR regime for $k \lesssim k_l \sim \mathcal{O}(10^{-3} k_\Lambda)$, where the gravitational couplings $G$ and $\Lambda$ tend to zero in the limit $k \to 0$ [6]; this scaling region can affect, however, the evolution of the Universe in the late future, which is out of the scope of our discussion in the present paper).

On the space of the dimensionless couplings ($g = Gk^2$, $\lambda = \Lambda/k^2$), the physical RG trajectory relevant for our Universe starts at the Reuter fixed point ($g_* \approx 0.707$, $\lambda_* \approx 0.193$). We note that the position of the Reuter fixed point depends slightly on the details of the RG analysis and can be influenced by the matter content of the early Universe, but our considerations make use of the existence of the Reuter fixed point rather than its position. For $k \approx k_\Lambda$, it approaches the Gaussian fixed point at ($g_G = 0, \lambda_G = 0$), and with the further decrease of the RG scale, $k$ runs away from the Gaussian fixed point towards positive values of $\lambda$, while the dimensionful couplings take their constant values observed in the present day. The scaling of the couplings slightly above the scale $k_\Lambda$ is the so-called perturbative regime. The proposed interpolation formulas recover the UV scaling laws and the constant values $G_0$ and $\Lambda_0$ below the scale $k_\Lambda$, and in the crossover regime $k \in [k_\Lambda, k_G]$ are motivated by the scaling in the perturbative regime. The interpolation formulas contain three free parameters, which are determined from the continuity of $G(k)$, $\Lambda(k)$, and that of the matter density $\rho^{(m)}$ at the dynamical scale $k_G$, while the scale $k_\Lambda \approx 8.2 \times 10^{-31} m_{\mathrm{Pl}}$ is taken from the RG analyses [22].

The interpolation formulas are applied to a rather simple model Universe making the assumptions that *(i)* the Universe is spatially homogeneous and isotropic, *(ii)* it is spatially flat, *(iii)* its matter content is assigned to a single type of barotropic fluid with the EOS $p^{(f)} = w\rho^{(f)}$ with the constant $0 \leq w < 1$ and $p^{(f)}$ and $\rho^{(f)}$ being its pressure and energy density, respectively, while *(iv)* the barotropic fluid is subjected to the law of local energy conservation separately. In this model, we identify the presence of the evolving cosmological coupling $\Lambda$ with that of the dark energy and call the quantity:

$$\rho^{(\Lambda)} = \frac{\Lambda}{8\pi G} \tag{1}$$

the density of the dark energy. It is rather giving a name to the $\Lambda$-component of the model; to go beyond the nature of dark energy is out of the scope of the present paper. Motivated by the interpretation used in [1], we take the point of view that $\rho^{(\Lambda)}$ can be considered as the field-independent potential energy density of a condensed scalar field, which implies the EOS $p^{(\Lambda)} = -\rho^{(\Lambda)}$ with the pressure $p^{(\Lambda)}$ of the dark energy. Since the constituent $\Lambda$ is represented in its own right explicitly in our model, we assumed that the equation of state parameter of the other constituent of matter is definitely larger than $w = -1$. The cases of barotropic fluids with a constant negative parameter $w \in (-1, 0)$, as well as those with $w \in (1/3, 1]$ are rather hypothetical ones according to our knowledge. For a one-component scalar field $\varphi(t, \vec{x})$ in the slow-roll inflationary scenario, the parameter $w = p^{(\varphi)}/\rho^{(\varphi)}$ takes initially negative values close to, but larger than $-1$ and may approach values either $w \approx 0$ (dust) or $w \approx 1/3$ (ultrarelativistic gas) at the graceful exit depending on the particular form of the inflaton potential [34].

It is well known that, if the Universe was filled by radiation or dust after the Big Bang, the quantum improvement may make unnecessary any fine-tuning of the initial conditions or the introduction of an inflaton field, in order to solve the horizon and flatness problems [29]. In that case, the physical phase trajectory runs to a universal attractor, being independent of the IR values of the gravitational couplings. The early time evolution

is characterized by the time-dependent Hubble parameter $H = h/t$ and the scale factor $a(t) = \mathcal{M}t^h$ with $h = \frac{4}{3(1+w)}$. For $h \geq 1$, both the flatness and horizon problems are solved automatically [29]: *(i)* the universal attractor starts from a singularity at $t = 0$, characterized by constant and equal cosmological parameters $\Omega^{(m)}$ and $\Omega^{(\Lambda)}$ of the matter and that of the dark energy, respectively, so that $\Omega^{(m)} = \Omega^{(\Lambda)} = \frac{1}{2}$ implies in total $\Omega^{(tot)} = 1$ and a flat spatial geometry; *(ii)* no finite particle horizon exists, so that our entire visible Universe should have been causally connected in the Planck era. It should be noted that the inequality $h \geq 1$ allows for considering hypothetical barotropic fluids with $-1 < w < 0$, as well. Our formulas found below remain valid for any values of the parameter $w$ in the interval $-1 < w < 1$.

Applying the proposed interpolation formulas to the above-described spatially flat model Universe enables one to solve analytically the problem of evolution for times from the Big Bang at $t = 0$ to the asymptotically far future $t \to \infty$. This circumstance enables one to obtain analytic expressions for the $k$-to-$t$ conversion rule in the Planck and crossover eras, as well as those for the characteristic time scales $t_G$ and $t_\Lambda$ corresponding to the dynamical RG scales $k_G$ and $k_\Lambda$, respectively. The discussion of the evolution problem is performed both in Scheme A and Scheme B. Both descriptions provide identical results for the entire cosmological evolution including the characteristic time scales. This is basically the consequence of the equivalence of the reduced consistency condition and the constraint on the RG parameters, Equations (14) and (24), respectively, in the case of the discussed particular model Universe. After continuously sewing the solutions on the various intervals for $k(t)$ at $t = t_G$ and those for $H(t)$ at $t = t_\Lambda$, the cosmological quantities $H(t)$, $\rho^{(f)}(t)$, and $\rho^{(\Lambda)}(t)$ become continuous. In the Planck era for $0 < t \leq t_G$, the well-known universal evolution governed by the Reuter fixed point is recovered with the simple conversion rule $k \propto 1/t$, while for late times $t \geq t_\Lambda$, the classical evolution holds trivially. In the crossover era for $t_G \leq t \leq t_\Lambda$, a nontrivial with increasing cosmological time strictly monotonically decreasing function $k(t)$ is obtained. Nevertheless, the simple proportionality $t_\Lambda \propto k_\Lambda$ remains yet a rather good approximation. The solution found in Scheme B enables us to identify the evolution in the time interval $0 < t \leq t_G$ as a cosmological fixed point with an evolving RG scale $k(t)$ and the dominance of a mixture of dark energy and barotropic fluid in equal proportions. The evolution of the model Universe in the late future for $t > t_\Lambda$ corresponds to another cosmological fixed point with a frozen-in RG scale at $k_\Lambda$ and dark energy domination in the asymptotic limit $t \to \infty$. It is argued that the evolution of dark energy is accompanied by the decrease of its entropy in the Planck and crossover eras, while the entropy of the barotropic fluid is preserved.

## 2. Interpolation Formulas

In this section, we propose simple formulas reflecting the main features of the RG flow of the gravitational couplings $G$ and $\Lambda$ in the four-dimensional, two-parameter EH gravity and interpolating between the UV and IR scaling regimes. In the parameter space of the dimensionless couplings $(g, \lambda)$, the RG trajectories are considered along which both of the couplings are nonnegative, running in the broken symmetry phase. The physically relevant trajectories emanate from the Reuter fixed point $(g_*, \lambda_*)$, pass the close neighborhood of the Gaussian fixed point, and possibly approach an IR fixed point for $k \to 0$. The latter is expressed in our treatment by saying that the couplings $G$ and $\Lambda$ take their present-day observed values in the IR region. This behavior, based on the results of much effort on the RG analysis of the EH theory of general relativity, was thoroughly discussed in [22]. There are two dynamical scales characterizing the RG flow: the scale $k_G$ above which the UV scaling laws $G = g_*/k^2$ and $\Lambda = \lambda_* k^2$ hold and the scale $k_\Lambda$ below which the IR regime with $G = G_0$ and $\Lambda = \Lambda_0$ is found. At the scale $k_\Lambda$, the RG trajectory is close to the Gaussian fixed point, so that for scales $k$ larger, but close to $k_\Lambda$, the perturbative scaling laws can be used (see, e.g., Equations (1.2) and (1.3) in [3], or Equation (26) in [22]). These perturbative laws motivated our choice of the interpolation formulas in the crossover regime $k \in [k_\Lambda, k_G]$. Thus, the following simple formulas are proposed:

$$G(k) \;=\; \begin{cases} G_0 & \text{for} \quad 0 \leq k \leq k_\Lambda \\ G_0 - b(k^2 - k_\Lambda^2) & \text{for} \quad k_\Lambda < k < k_G \\ g_* k^{-2} & \text{for} \qquad k_G \leq k \end{cases} \tag{2}$$

$$\Lambda(k) \;=\; \begin{cases} \Lambda_0 & \text{for} \quad 0 \leq k < k_\Lambda \\ \Lambda_0 + c(k^4 - k_\Lambda^4) & \text{for} \quad k_\Lambda \leq k < k_G \\ \lambda_* k^2 & \text{for} \qquad k_G \leq k \end{cases} \tag{3}$$

where $k_\Lambda \approx 8.2 \times 10^{-31} m_{\text{Pl}}$ is taken from the RG analysis [22] and $k_G$, $b$, $c$ are still free parameters. By construction, the formulas are continuous at the scale $k = k_\Lambda$. Two algebraic equations for the determination of the three yet-unknown parameters are the continuity conditions for $G$ and $\Lambda$ at the scale $k_G$. The third equation is obtained by requiring the continuity of the ratio of the time derivatives of the gravitational couplings $r_{\dot\Lambda/\dot G} \equiv \frac{d\Lambda(k)/dk}{dG/dk}$ at the scale $k_G$. (Here and below, the dot denotes the time derivative.) Since local energy conservation is supposed to hold separately for matter, the consistency condition $\nabla_\mu(GT^{\mu\nu}) = 0$ reduces to an expression, the reduced consistency condition (cf. Equation (14) below) according to which the energy density of matter is proportional to the ratio $r_{\dot\Lambda/\dot G}$. Therefore, the requirement of the continuity of that ratio ensures the continuity of matter density $\rho^{(m)}$ at the scale $k_G$. We shall see that the three requirements settle $k_G$ at a value of the correct order of magnitude, $k_G \sim \mathcal{O}(m_{Pl})$. The continuity conditions are then the following:

$$\Lambda(k_G) \;=\; \Lambda_0 + c(k_G^4 - k_\Lambda^4) = \lambda_* k_G^2, \tag{4}$$

$$G(k_G) \;=\; G_0 - b(k_G^2 - k_\Lambda^2) = \frac{g_*}{k_G^2}, \tag{5}$$

$$r_{\dot\Lambda/\dot G}(k_G) \;=\; -\frac{2ck_G^2}{b} = -\frac{\lambda_*}{g_*}k_G^4. \tag{6}$$

Making the working hypotheses that the order-of-magnitude estimates $k_G \sim m_{Pl} \gg k_\Lambda$ and $ck_G^4 \gg \Lambda_0$ hold, one obtains for the free parameters of the interpolation formulas:

$$k_G^2 = \frac{3g_*}{G_0}, \quad b = \frac{2}{9g_*}G_0^2, \quad c = \frac{\lambda_*}{3g_*}G_0. \tag{7}$$

Then, we see that our working hypotheses are rather well-satisfied, because the following estimates can be made: $k_G = \sqrt{3g_*}\, m_{Pl} \approx 1.46 m_{Pl}$ and $ck_G^4 = \lambda_* k_G^2 = \frac{3g_*\lambda_*}{G_0} \approx 0.4 m_{Pl}^2 \gg \Lambda_0 \approx 3 \cdot 10^{-122} m_{Pl}^2$.

One can check that the interpolation Formula (2) provides a good approximation of the IR scaling law for scales close to, but above the scale $k_\Lambda$ given by Equation (26) in [22]. The latter can be approximated for $k \gtrsim k_\Lambda$, i.e., for $k - k_\Lambda \ll k_G$ by

$$G(k) \;\approx\; G_0\left(1 - \frac{7}{6\pi}G_0(k^2 - k_\Lambda^2)\right), \tag{8}$$

where $G(k_\Lambda) = G_0$ was used. A comparison of the estimate (8) with our interpolation Formula (2) yields $b^{(est)} = \frac{7}{6\pi}G_0^2 \approx 0.37 G_0^2$, while according to our setting above, $b = \frac{2}{9g_*}G_0^2 \approx 0.31 G_0^2$.

Finally, it was checked numerically that the interpolation Formulas (2) and (3) with the parameter setting (7) reproduce fairly well the RG evolution of the couplings $G$ and $\Lambda$ over the entire range of the RG scale covering roughly 40 orders of magnitude (see Figure 4 in [22]).

### 3. Model Universe

In this section, we briefly overview the equations governing the evolution of the model Universe with the properties *(i)–(iv)*. Below, we intend to follow the evolution of this simple model in Schemes A and B independently.

#### 3.1. Scheme A

In order to be somewhat more general, let us consider models with the properties *(i) and (ii)* containing some matter with energy density $\rho^{(m)}$ and pressure $p^{(m)}$. Assuming the adiabatic expansion of matter implies the law of local energy conservation:

$$\dot{\rho}^{(m)} + 3H(\rho^{(m)} + p^{(m)}) \quad = \quad 0, \tag{9}$$

where $H = \dot{a}/a$ is the Hubble parameter and $a$ denotes the scale factor of the Universe. The local energy conservation for matter does not involve the gravitational couplings explicitly, so that it keeps its unimproved form. Nevertheless, the time-dependence of the Hubble parameter $H$ is affected by quantum effects. The quantum-improved Friedmann equations are given via the replacement of the observed constants $G_0$ and $\Lambda_0$ by their time-dependent counterparts $G(t)$ and $\Lambda(t)$:

$$H^2 \quad = \quad \frac{8\pi G(t)}{3}\rho(t) = \frac{8\pi G(t)}{3}\rho^{(m)} + \frac{\Lambda(t)}{3}, \tag{10}$$

$$\frac{\ddot{a}}{a} \quad = \quad -\frac{4\pi G(t)}{3}(\rho^{(m)} + 3p^{(m)}) + \frac{\Lambda(t)}{3}. \tag{11}$$

Equation (11) can be replaced by the simpler equation:

$$\dot{H} \quad = \quad -4\pi G(t)(\rho + p) = -4\pi G(t)(\rho^{(m)} + p^{(m)}), \tag{12}$$

where $\rho = \rho^{(m)} + \rho^{(\Lambda)}$ and $p = p^{(m)} + p^{(\Lambda)}$ stand for the total energy density and pressure, respectively. The critical density $\rho_c = H^2/(8\pi G(t))$ is identical to the total energy density, since we restricted ourselves to models with flat spatial geometry. Equations (11) and (12) are dynamical equations, whereas Equation (10) is the so-called Friedmann constraint laid on the cosmological variables. Equations (10) and (11) imply the consistency condition:

$$\dot{\rho}^{(m)} + 3H(\rho^{(m)} + p^{(m)}) \quad = \quad -\frac{8\pi\rho^{(m)}\dot{G} + \dot{\Lambda}}{8\pi G} \equiv \mathcal{P}^{(m)}, \tag{13}$$

which is the symmetry-reduced form of the condition $\nabla_\mu(GT^{\mu\nu}) = 0$. Combining the latter with the local energy conservation for matter, one finds the reduced consistency condition:

$$8\pi\rho^{(m)}\dot{G} + \dot{\Lambda} \quad = \quad 0, \tag{14}$$

i.e., the vanishing of the quantity $\mathcal{P}^{(m)}$. The condition $\mathcal{P}^{(m)} = 0$ expresses that, during the evolution, the entropy of the matter is conserved [3]. Assuming the consistency condition (13), as well as the law of the local energy conservation of matter is in agreement with the treatment used in [28–30].

Although our interpolation formulas are not differentiable at $t = t_G$ and $t = t_\Lambda$, the continuity of the matter density $\rho^{(m)}$ ensures the continuity of the Hubble parameter $H$ and that of its first time derivative via the Friedmann Equations (10) and (12), respectively, which implies the continuity of the scale factor $a$ and its first and second time derivatives. Furthermore, the continuity of the matter density $\rho^{(m)}$ implies that of its first time derivative through Equation (13) with $\mathcal{P}^{(m)} = 0$.

It is noteworthy to make here a note on the entropy of the dark energy. In the case of time-dependent $G$ and $\Lambda$, the equation holds:

$$\dot{\rho}^{(\Lambda)} + 3H(\rho^{(\Lambda)} + p^{(\Lambda)}) \quad = \quad \dot{\rho}^{(\Lambda)} = \mathcal{P}^{(\Lambda)} \tag{15}$$

with

$$\mathcal{P}^{(\Lambda)} = \frac{\dot{\Lambda} - 8\pi\rho^{(\Lambda)}\dot{G}}{8\pi G} = -\rho_c\frac{\dot{G}}{G} \neq 0, \tag{16}$$

where we made use of the EOS of the dark energy and the relation $\rho^{(\Lambda)} = \rho - \rho^{(m)} = \rho_c - \rho^{(m)}$. Multiplying Equation (15) by the proper volume $\mathcal{V}$ of the Universe and comparing it by the first law of thermodynamics applied to the dark energy as in [3], one reads off from Equation (16) that the entropy $S^{(\Lambda)}$ of the dark energy changes, $T\dot{S}^{(\Lambda)} = \mathcal{P}^{(\Lambda)}\mathcal{V}$, where $T$ is the temperature of the dark energy. As a rule, the inequalities $dG(k)/dk < 0$ and $\dot{k} < 0$ hold, implying that $\dot{G} = \dot{k}(dG/dk) > 0$, so that $\mathcal{P}^{(\Lambda)} < 0$. Thus, the entropy $S^{(\Lambda)}$ should decrease in a realistic case. This does not contradict the second law of thermodynamics, because the dark energy "in the auxiliary field $G(t)$" does not represent a closed thermodynamical system. Assuming that the matter and the dark energy are in thermodynamical equilibrium, the temperature $T$ can be estimated by making use of the Stefan–Boltzmann law, $T = (\rho^{(m)}/\sigma)^{1/4}$ with the constant $\sigma$ determined by the number of independent degrees of freedom of matter. Then, we obtain the estimate:

$$\dot{S}^{(\Lambda)} = -\mathcal{V}\sigma^{1/4}\frac{\dot{G}}{G}\frac{\rho_c}{(\rho^{(m)})^{1/4}}. \tag{17}$$

The above-listed equations incorporate implicitly the yet-undetermined function $k = k(t)$, the $k$-to-$t$ conversion rule. In the case of cosmological expansion, it is expected that the RG scale $k$ can be uniquely converted into the cosmological time $t$ in such a manner that strictly monotonically decreasing $k$ corresponds to continuously and strictly monotonically increasing $t$. Therefore, characteristic times $t_G < t_\Lambda$ can be assigned to the dynamical RG scales $k_\Lambda < k_G$. Let us turn now to the particular model characterized by the properties (i)–(iv) and described in terms of the cosmological variables $H(t)$, $\rho^{(m)}(t) = \rho^{(f)}(t)$ and the conversion rule $k = k(t)$. For $k < k_\Lambda$, i.e., for $t > t_\Lambda$, the couplings $G$ and $\Lambda$ are frozen at their values $G_0$ and $\Lambda_0$ taken at the scale $k_\Lambda$, the consistency condition (13) reduces to the law of the local energy conservation of matter, i.e., to Equation (9), so that we have two independent equations, say Equations (9) and (10) for the two variables $H(t)$ and $\rho^{(f)}(t)$ for times $t > t_\Lambda$. Therefore, the classical evolution takes place for times $t > t_\Lambda$ with the initial conditions given at $t_\Lambda$. For times $0 < t \leq t_\Lambda$, quantum effects come into play via the evolving gravitational couplings, and we have one more variable, $k(t)$, as well as one more equation, the reduced consistency condition (14). We take the point of view of [31] that, for a given RG flow $G(k)$ and $\Lambda(k)$, the reduced consistency condition (14) can be interpreted as an implicit equation for the determination of the conversion rule $k = k(t)$.

### 3.2. Scheme B

Now, we turn to the description of the model with the properties (i)–(iv) in terms of the dimensionless cosmological variables proposed in [1,32,33]. That model is the particular case of the models discussed in [1], although in our model, the scalar field $\varphi$ with the potential energy density $V(\varphi)$ is degenerated in the sense that it exhibits a single degree of freedom incorporated in the actual value of $\Lambda$. Namely, we can think of the potential $V(\varphi) \equiv \rho^{(\Lambda)}$ being independent of the field variable $\varphi$ and the field variable to be frozen at any constant value, since a field-independent potential has a continuum of minimum places. Below, we apply the description given in [1] to our particular model. In such an approach, the dimensionless variables are

$$x = \pm\sqrt{\frac{\dot{\varphi}^2/2}{\rho_c}} \equiv 0, \quad y = \sqrt{\frac{V}{\rho_c}} = \frac{\sqrt{\Lambda}}{\sqrt{3}H}, \tag{18}$$

$$z = \frac{V_{,\varphi}(\varphi)}{8\pi G V(\varphi)} \equiv 0, \quad \Omega^{(f)} = \frac{\rho^{(f)}}{\rho_c} \tag{19}$$

with $V_{,\varphi} = \partial V / \partial \varphi$, and also the quantity $\eta = V_{,\varphi\varphi} / (8\pi G V) = 0$ vanishes identically. For later use, we shall introduce the cosmological parameters $\Omega^{(\Lambda)} = \rho^{(\Lambda)} / \rho_c = y^2$ and $\Omega = \Omega^{(f)} + \Omega^{(\Lambda)}$. Instead of the cosmological time, it is more convenient to use the dimensionless evolution parameter $N = -\ln a$ related to the time variable $t$ via the relation $H dt = -dN$. Therefore, the $k$-to-$t$ conversion rule reappears through the yet-unknown function $k = k(N)$. A general field-dependent potential, as well as the position of its minimum exhibit scale dependences, which should be invoked from the RG analysis of the scalar field coupled to gravity. Nevertheless, in our model, the only scale dependence of the minimum value of the potential energy density is already incorporated in the scale dependence of $\rho^{(\Lambda)}$, i.e., in that of the gravitational couplings $G$ and $\Lambda$. The RG scale dependences are characterized by the beta functions given in [1]:

$$\eta_{\mathrm{RG}} = \frac{\partial \ln G}{\partial \ln k}, \quad \nu_{\mathrm{RG}} = \frac{\partial \ln \Lambda}{\partial \ln k} - \eta_{\mathrm{RG}}, \tag{20}$$

which are independent of the field variable $\varphi$ in our case, and the quantity $\sigma_{\mathrm{RG}}$ falls off trivially. The law of the local energy conservation (9) applied to the barotropic fluid (cf. Equation (22) in [1]) is rewritten as

$$\frac{d\Omega^{(f)}}{dN} = 3\gamma\Omega^{(f)}(1 - \Omega^{(f)}) + \Omega\eta_{\mathrm{RG}}\frac{d\ln k}{dN}. \tag{21}$$

Equations (19) and (21) in [1] for the evolution of $x$ and $z$ turn to the trivial identity $0 = 0$, while Equation (20) in [1] is rewritten as

$$\frac{dy}{dN} = y\left(-\frac{3}{2}\gamma\Omega^{(f)} + \frac{1}{2}(\eta_{\mathrm{RG}} + \nu_{\mathrm{RG}})\frac{d\ln k}{dN}\right). \tag{22}$$

The Friedmann constraint (10) is rewritten as

$$0 = 1 - y^2 - \Omega^{(f)} \tag{23}$$

and the constraint on the beta functions given by Equation (25) in [1] yields

$$0 = \eta_{\mathrm{RG}}(k) + y^2 \nu_{\mathrm{RG}}(k). \tag{24}$$

In the particular model discussed here, the constraint (24) is equivalent to the reduced consistency condition (14), since the $t$-dependence of the gravitational couplings occurs implicitly via their $k$-dependences, and (14) can be rewritten as

$$-0 = 8\pi\rho^{(f)}\eta_{\mathrm{RG}} + \frac{\Lambda}{G}(\eta_{\mathrm{RG}} + \nu_{\mathrm{RG}}). \tag{25}$$

Dividing both sides of this equation by $8\pi\rho_c$, we obtain

$$\begin{aligned} 0 &= \Omega^{(f)}\eta_{\mathrm{RG}} + \Omega^{(\Lambda)}(\eta_{\mathrm{RG}} + \nu_{\mathrm{RG}}) \\ &= (\Omega^{(f)} + y^2)\eta_{\mathrm{RG}} + y^2 \nu_{\mathrm{RG}}, \end{aligned} \tag{26}$$

which is identical to the constraint (24) since the relation $y^2 = \Omega^{(\Lambda)}$ and the Friedmann constraint (23) holdin our case.

Now, the dynamical Equation (27) in [1] gives

$$\frac{d\ln k}{dN} = \frac{3}{2\alpha_{\mathrm{RG}}}\gamma\Omega^{(f)} \tag{27}$$

with

$$\alpha_{\mathrm{RG}} = \frac{1}{2}\left[(\eta_{\mathrm{RG}} + \nu_{\mathrm{RG}}) - \frac{d}{d\ln k}\ln\left(-\frac{\eta_{\mathrm{RG}}}{\nu_{\mathrm{RG}}}\right)\right] \tag{28}$$

in our case. In this description, the quantum improvement is taken into account in the dynamical Equations (21) and (22) by the terms proportional to $d\ln k/dN$ and the dynamical Equation (27). In the classical case, the evolving scale $k = k(N)$ and the terms proportional to $d\ln k/dN$ fall off. Due to the Friedmann constraint (23), the classical model exhibits a single independent variable (e.g., $y$), while the quantum-improved one exhibits two independent variables (e.g., $y$ and $k$). For the later discussion, the relation of the flow of the Hubble parameter to that of the RG scale $k$ given by Equation (28) in [1] becomes useful, which is rewritten as

$$\frac{d\ln H}{dN} = \alpha_{\mathrm{RG}}\frac{d\ln k}{dN} \tag{29}$$

in our case.

## 4. Analytical Solution

In this section, we show that both Schemes A and B provide the same analytical result for the evolution of the discussed model Universe along the entire time axis. First, the analytical solutions were determined in the time intervals $[0, t_G]$, $[t_G, t_\Lambda]$, and $[t_\Lambda, \infty)$ separately, and then, these sectional solutions were sewn continuously at the times $t_G$ and $t_\Lambda$ requiring the continuity of $k(t)$ at $t = t_G$ and that of $H(t)$ at $t = t_\Lambda$. The classical solution valid for $t_\Lambda \le t < \infty$ was found in Scheme B and the constant of integration in $H(t)$ settled by its continuity at $t = t_\Lambda$. The zero of the cosmological time is chosen when the expanding solution has its initial singularity.

Independent of the scheme used, the local energy conservation (9) for the barotropic fluid yields the relation:

$$\rho^{(f)} = \mathcal{M}a^{-3\gamma}, \tag{30}$$

where the (dimensionful) constant of integration $\mathcal{M}$ is a free parameter, determining the matter content of the model Universe. The relation (30) holds for all times $0 < t < \infty$, so that $\mathcal{M}$ remains the only free parameter after the solution in the various time intervals has been sewn continuously.

### 4.1. Scheme A

Applying the independent Equations (9), (10) and (14) to our model for $0 < t \le t_\Lambda$, Equations (14) and (9) enable one to express the variables $\rho^{(f)}(t)$ and $H(t)$ in terms of the function $k(t)$:

$$\rho^{(f)} = -\frac{1}{8\pi}\frac{\dot\Lambda}{\dot G} = \begin{cases} \frac{\lambda_*}{8\pi g_*}k^4 & \text{for} \quad 0 < t \le t_G, \\ \frac{3\lambda_*}{8\pi G_0}k^2 & \text{for} \quad t_G \le t \le t_\Lambda \end{cases} \tag{31}$$

$$H = -\frac{2}{3\gamma}\frac{\dot k}{k}\begin{cases} 2 & \text{for} \quad 0 < t \le t_G, \\ 1 & \text{for} \quad t_G \le t \le t_\Lambda \end{cases} \tag{32}$$

respectively, where we made use of the scale dependences given in Equations (2) and (3). Inserting the expressions (31) and (32) into the Friedmann constraint (10), one obtains a first-order ordinary differential equation (ODE) for the function $k(t)$, the explicit form of which depends on the explicit scale dependence of the gravitational couplings given in Equations (2) and (3). In this manner, we see that the dynamics contains a single additional constant of integration in both time intervals $[0, t_G]$ and $[t_G, t_\Lambda]$ occurring when the ODE for the function $k(t)$ is integrated.

### 4.1.1. Time Interval $0 < t \leq t_G$

The ODE for $k(t)$ takes the form:

$$\dot{k} = -\sqrt{\frac{3\lambda_*}{8}}\gamma k^2, \tag{33}$$

where we consider the case with the negative square root the realistic one for the expanding Universe, since it yields increasing time $t$ with decreasing RG scale $k$. The solution of Equation (33) is given as

$$k(t) = \sqrt{\frac{8}{3\lambda_*}}\frac{1}{\gamma t}, \tag{34}$$

where we shifted the singularity of the solution to $t = 0$ by the appropriate choice of the constant of integration. Inserting this back into Equations (31), (32), (2), and (3), the time dependences of all interesting cosmological quantities can be made explicit:

$$H(t) = \frac{4}{3\gamma t} \equiv \frac{h}{t}, \tag{35}$$

$$\rho^{(f)}(t) = \frac{8}{9\pi g_* \lambda_* \gamma^4 t^4} = \rho^{(\Lambda)} = \frac{1}{2}\rho_c, \tag{36}$$

$$\rho_c(t) = \frac{3H^2}{8\pi G} = \frac{16}{9\pi g_* \lambda_* \gamma^4 t^4}, \tag{37}$$

$$\Omega^{(f)} = \Omega^{(\Lambda)} = \frac{1}{2}. \tag{38}$$

The comparison of Equations (34) and (35) reveals that the simple proportionality holds:

$$k = \sqrt{\frac{3}{2\lambda_*}}H \tag{39}$$

in the time interval $0 < t \leq t_G$. The inverse proportionality of the Hubble parameter to the time $t$ leads to the time dependence of the scale parameter:

$$a = At^h, \tag{40}$$

and the comparison of Equation (30) with (36) gives

$$A = \left(\frac{9\pi g_* \lambda_* \gamma^4}{8}\mathcal{M}\right)^{\frac{1}{3\gamma}}. \tag{41}$$

We see that the usage of the reduced consistency condition (14) has led to the naive inverse proportionality $k \propto 1/t \propto H$ of the RG scale $k$ to the cosmological time $t$ in the interval $0 < t \leq t_G$. The cosmological evolution is unique in this interval for any given model characterized with a given value of the parameter $\mathcal{M}$. Evaluating the right-hand side of the expression (16), it is straightforward to check that the equality holds:

$$\mathcal{P}^{(\Lambda)} = -\frac{32}{9\pi g_* \lambda_* \gamma^4 t^5} = \dot{\rho}^{(\Lambda)}. \tag{42}$$

Furthermore, one obtains from Equation (17) that $\dot{S}^{(\Lambda)} \propto -1/t^4$, i.e., the rate of the negative entropy production falls off rapidly during the Planck era $0 < t \leq t_G$.

Since the solution is uniquely settled in the interval $[0, t_G]$, one is enabled to determine the characteristic time scale corresponding to the dynamical RG scale $k_G$ through the requirement $k(t_G - 0) = k_G$:

$$t_G = \sqrt{\frac{8}{3\lambda_*}} \frac{1}{\gamma k_G} = \sqrt{\frac{8}{9g_*\lambda_*\gamma^2}} m_{Pl}^{-1},$$

(43)

where we made use of Equations (7) and (34). Thus, we obtained that $t_G \propto 1/k_G$ in the discussed model.

4.1.2. Time Interval $t_G \leq t \leq t_\Lambda$

For later use, we rewrite the RG scale dependences (2) and (3) of the gravitational couplings as

$$G(k) = b\left(-k^2 + \frac{1}{2}E\right), \quad \Lambda(k) = c(k^4 + F)$$

(44)

in the interval $k \in [k_\Lambda, k_G]$, where

$$E = 2\left(\frac{G_0}{b} + k_\Lambda^2\right) = 3k_G^2\left(1 + \frac{2}{3}\frac{k_\Lambda^2}{k_G^2}\right),$$

(45)

$$F = \frac{\Lambda_0}{c} - k_\Lambda^4 = k_G^4\left(\frac{G_0\Lambda_0}{3g_*\lambda_*} - \frac{k_\Lambda^4}{k_G^4}\right) \approx -8.4 \times 10^{-120}k_G^4.$$

(46)

Taking again the negative square root, the dynamical equation for the RG scale $k = k(t)$ can now be recast as

$$\dot{k} = -\frac{3\gamma}{2}kD^{1/2},$$

(47)

with

$$D = \frac{\lambda_*G(k)k^2}{G_0} + \frac{\Lambda(k)}{3} = -b_2u^2 + b_1u + b_0,$$

(48)

where the variable $u = k^2$ and the coefficients:

$$b_0 = \frac{\Lambda_0}{3} - \frac{\lambda_*G_0}{9g_*}k_\Lambda^4 \approx -0.47 \times 10^{-122}m_{Pl}^2,$$

(49)

$$b_1 = \lambda_*\left(1 + \frac{2G_0}{9g_*}k_\Lambda^2\right) \approx \lambda_*[1 + \mathcal{O}(k_\Lambda^2/k_G^2)],$$

(50)

$$b_2 = \frac{\lambda_*}{9g_*}m_{Pl}^2$$

(51)

were introduced. The quantity $D$ is definitively positive by construction in the interval $k \in [k_\Lambda, k_G]$. Multiplying both sides of Equation (47) by $2k$, one can rewrite it as

$$\dot{u} = -3\gamma uD^{1/2}(u).$$

(52)

The inverse function $t = t(u)$ is then given as

$$t - t_2 = -\frac{1}{3\gamma}I(u)$$

(53)

with the constant of integration $t_2$ and the integral

$$I(u) = \int \frac{du}{u D^{1/2}(u)}. \tag{54}$$

Since the coefficient $b_0 < 0$ and the discriminant of the quadratic form $D(u)$ is $\Delta = -4b_0 b_2 - b_1^2 \approx -\lambda_*^2[1 + \mathcal{O}(k_\Lambda^2/k_G^2)] < 0$, one obtains (see Item 2.266 in [35])

$$
\begin{aligned}
I(u) &= (-b_0)^{-1/2} \arctan \mathcal{I}(u) \text{ with} \\
\mathcal{I}(u) &= \frac{2b_0 + b_1 u}{2[-b_0 D(u)]^{1/2}}
\end{aligned}
\tag{55}
$$

and rewrites Equation (53) as

$$\mathcal{I}(u) = \mathcal{I}(t) \equiv \tan\left(-3\gamma\sqrt{-b_0}(t - t_2)\right). \tag{56}$$

Since $b_1 \approx \lambda_*$ with the accuracy of 60 valuable digits, this yields the second-order algebraic equation for $u$:

$$0 \approx [\lambda_*^2 - 4b_0 b_2 \mathcal{I}^2(t)]u^2 + 4b_0(\lambda_* u + b_0)[1 + \mathcal{I}^2(t)] \tag{57}$$

having the roots:

$$u_\pm = \frac{2b_0}{\lambda_*}\frac{-1 \pm \sqrt{1 - \Xi(t)}}{\Xi(t)} \tag{58}$$

with

$$\Xi(t) = \frac{\lambda_*^2 - 4b_0 b_2 \mathcal{I}^2(t)}{\lambda_*^2[1 + \mathcal{I}^2(t)]} > 0. \tag{59}$$

Below, we show that only the root $u_-$ is physical. Making use of Equation (43), the constant of integration $t_2$ is settled by the relation $I(k_G^2) = -3\gamma(t_G - t_2)$, which yields

$$t_2 = t_G + \frac{1}{3\gamma\sqrt{-b_0}}\arctan \mathcal{I}(k_G^2). \tag{60}$$

The estimate of $\mathcal{I}(k_G^2)$ with an accuracy of 60 valuable digits gives $10^{61}$. For a huge number $\mathcal{I}$, it is a good approximation to take

$$\arctan \mathcal{I} \approx \frac{\pi}{2} - \frac{1}{\mathcal{I}}. \tag{61}$$

This estimate enables one to determine the second term on the right-hand side of Equation (60) with an accuracy of the order $\mathcal{O}(t_G/t_2)$ to obtain

$$t_2 = \frac{2}{3}t_G + \frac{\pi}{6\gamma\sqrt{-b_0}} \tag{62}$$

and hence,

$$\mathcal{I}(t) = \cotan\left[3\gamma\sqrt{-b_0}\left(t - \frac{2}{3}t_G\right)\right]. \tag{63}$$

The next relation $\mathcal{I}(k_\Lambda^2) = \mathcal{I}(t_\Lambda)$ enables one to determine the time scale $t_\Lambda$ corresponding to the dynamical scale $k_\Lambda$. Here, $\mathcal{I}(k_\Lambda^2) \approx 2.6 \times 10^{30}$ is also a huge number, so that one makes use of the estimate (61) once more to obtain

$$t_\Lambda = \frac{2}{3}t_G + \frac{2}{3\gamma\sqrt{\lambda_*}k_\Lambda}. \tag{64}$$

This means that, with an accuracy of 30 valuable digits, $t_\Lambda$ is inversely proportional to $k_\Lambda$. The knowledge of $t_\Lambda$ enables one to decide that only the root $u_-$ reproduces $k_\Lambda^2$ when the expression (64) for $t_\Lambda$ is inserted into the right-hand side of Equation (58). It is worthwhile mentioning that $\Xi(t_\Lambda) \approx 10^{-60}$ and $\Xi(t_G) \approx 10^{-122}$, so that $\Xi$ remains very small as compared to 1 in the entire interval $[t_G, t_\Lambda]$. This justifies the approximation $\sqrt{1-\Xi} \approx 1 - \frac{1}{2}\Xi$. Therefore, the root $u_-$ of (58) provides with rather high accuracy that

$$k(t) \approx \sqrt{\frac{-4b_0}{\lambda_*\Xi}} \approx \sqrt{3}k_G(1 + B_H\bar{t}^2)^{-1/2} \tag{65}$$

with $B_H = \frac{81g_*\lambda_*\gamma^2}{4}m_{\text{Pl}}^2$ and the shorthand notation $\bar{t} = t - \frac{2}{3}t_G$. It is straightforward to check that, even with this estimated form of the conversion rule $k = k(t)$, the equality $k(t_G) = k_G$ holds.

The comparison of Equation (32) with Equation (47) yields

$$H = \sqrt{D}, \tag{66}$$

a rephrasing of the Friedmann constraint. Therefore, the relations (31):

$$\rho^{(\Lambda)} = \frac{c(k^4 + F)}{8\pi b(-k^2 + \frac{1}{2}E)}, \quad \rho_c = \frac{3D}{8\pi b(-k^2 + \frac{1}{2}E)}, \tag{67}$$

imply that $\rho_c = \rho^{(f)} + \rho^{(\Lambda)}$ and $\Omega^{(f)} + \Omega^{(\Lambda)} = 1$, which is another form of the Friedmann constraint. Equations (66) and (48) reveal a nontrivial relationship between the RG scale $k$ and the Hubble parameter $H$ in the time interval $t_G \leq t \leq t_\Lambda$ instead of the simple proportionality $k \propto H$ valid in the interval $0 \leq t \leq t_G$. For later discussions, it is useful to rewrite (47) in the form:

$$\frac{d\ln k}{dt} = -\frac{3\gamma}{2}H(k). \tag{68}$$

It is important to notice that both Equations (66) and (68) are exact equations.

Since the expression (65) is an estimate, we may obtain somewhat different results for the function $H(t)$ when evaluating it in various ways:

$$H^{(1)}(t) = \sqrt{D(k^2(t))} \text{ or } H^{(2)}(t) = -\frac{2\dot{k}}{3\gamma k}, \tag{69}$$

although $H^{(1)}(t)$ and $H^{(2)}(t)$ are identical for the exact form of the function $k(t)$. The nonvanishing of the quantity:

$$\delta_H(t) = \frac{H^{(2)}(t) - H^{(1)}(t)}{H^{(2)}(t)} \tag{70}$$

characterizes the accuracy of the usage of the estimate (65). A straightforward, but lengthy evaluation yields

$$H^{(1)}(t) = 3\sqrt{g_*\lambda_*}\, m_{\text{Pl}} \left[ -\frac{1}{(1+B_H \bar{t}^2)^2} + \frac{1}{1+B_H \bar{t}^2}\left(1 + \frac{2k_\Lambda^2}{3k_G^2}\right) + \frac{F}{9k_G^4} \right]^{1/2}, \quad (71)$$

$$H^{(2)}(t) = \frac{2}{3\gamma}\frac{B_H \bar{t}}{1+B_H \bar{t}^2}, \quad (72)$$

$$\delta_H(t) = 1 - \left[ 1 + \frac{2k_\Lambda^2}{3k_G^2} + \frac{1}{B_H \bar{t}^2}\frac{2k_\Lambda^2}{3k_G^2} + \frac{(1+B_H \bar{t}^2)^2}{B_H \bar{t}^2}\frac{F}{9k_G^4} \right]^{1/2} \equiv 1 - A_H^{1/2}(t). \quad (73)$$

The numerical values of $\delta_H(t)$ are extremely small, e.g., one obtains $\delta_H(t_G) \approx -\frac{1}{2}\frac{k_\Lambda^2}{k_G^2} \sim \mathcal{O}(10^{-60})$ and $\delta_H(t_\Lambda) \approx \frac{1}{6}\frac{k_\Lambda^2}{k_G^2} - \frac{\Lambda_0}{2\lambda_* k_\Lambda^2} \sim \mathcal{O}(10^{-60})$. Thus, $H^{(1)}(t)$ and $H^{(2)}(t)$ provide identical numerical results with the accuracy of 60 valuable digits. With the help of the estimate (65), it is now straightforward to obtain the time dependences of the other interesting cosmological quantities. In order not to violate the Friedmann constraint by the time-dependent analytical expressions, we evaluate the critical density by making use of $H^{(1)}(t)$ given in Equation (71). Then, we obtain

$$\rho^{(f)}(t) = \frac{3\lambda_* m_{\text{Pl}}^2 k^2}{8\pi} = \frac{B_H m_{\text{Pl}}^2}{6\pi\gamma^2(1+B_H \bar{t}^2)}, \quad (74)$$

$$\rho_c(t) = \frac{3[H^{(1)}(t)]^2}{8\pi G(t)} = \frac{27 g_* \lambda_* m_{\text{Pl}}^2 A_H(t)}{8\pi b[-k^2(t) + \frac{1}{2}E]}, \quad (75)$$

$$\Omega^{(f)}(t) = \frac{\rho^{(f)}(t)}{\rho_c(t)} = \frac{1}{A_H(t)}\left[ -\frac{2}{(1+B_H \bar{t}^2)^2} + \frac{1}{1+B_H \bar{t}^2}\left(1 + \frac{2k_\Lambda^2}{3k_G^2}\right) \right], \quad (76)$$

$$\Omega^{(\Lambda)}(t) = \frac{\Lambda(t)}{3[H^{(1)}(t)]^2} = \frac{1}{A_H(t)}\left[ \frac{1}{(1+B_H \bar{t}^2)^2} + \frac{F}{9k_G^4} \right]. \quad (77)$$

Now, it is trivial that adding Equations (76) and (77) recapitulates the identity $\Omega^{(f)} + \Omega^{(\Lambda)} = 1$. The time dependence of the Hubble parameter (72) implies the time dependence of the scale factor:

$$a(t) = C_a(1+B_H \bar{t}^2)^{\frac{1}{3\gamma}}, \quad (78)$$

where the constant of integration $C_a = \left(\frac{8\pi}{27 g_* \lambda_*}\mathcal{M}\right)^{\frac{1}{3\gamma}} m_{\text{Pl}}^{-\frac{4}{3\gamma}}$ is set by the continuity of $a(t)$ at $t = t_G$. The comparison of Equations (65) and (78) yields the relation:

$$k = \sqrt{\frac{8\pi}{3\lambda_* m_{\text{Pl}}^2}}\mathcal{M} a^{-\frac{3\gamma}{2}}. \quad (79)$$

The quantity $-\mathcal{P}^{(\Lambda)}$ characterizing the entropy production is

$$\mathcal{P}^{(\Lambda)}(t) = -\rho_c\frac{\dot{G}}{G} = -\frac{81 g_* \gamma m_{\text{Pl}}^4}{16\pi}\frac{k^2(t)D^{3/2}[k^2(t)]}{(-k^2(t)+\frac{1}{2}E)^2}, \quad (80)$$

where one has to insert the function $k(t)$ from Equation (65) into the right-hand side. Making use of Equation (17), one can take the ratio of the rates of entropy production at the end $t_\Lambda$ and at the beginning $t_G$ of the crossover era. A somewhat lengthy, but straightforward evaluation yields the estimate:

$$\frac{\dot{S}^{(\Lambda)}|_{t=t_\Lambda}}{\dot{S}^{(\Lambda)}|_{t=t_G}} \approx \frac{\bar{t}_\Lambda}{\bar{t}_G}\left(\frac{k_\Lambda}{k_G}\right)^{7/2}[1 + \mathcal{O}(k_\Lambda^2/k_G^2)] \approx 10^{-75}. \quad (81)$$

This means that the entropy production rate radically falls off during the crossover era.

It is straightforward to check that the cosmological quantities we are interested in are all continuous at $t = t_G$ at least with the accuracy of 60 valuable digits corresponding to the ratio $k_\Lambda^2/k_G^2$. Equation (64) shows that the characteristic time scale $t_\Lambda$ corresponding to the RG scale $k_\Lambda$ is equal to $\bar{t}_\Lambda$ with the accuracy of $t_G/t_\Lambda \sim k_\Lambda/k_G \sim 10^{-30}$, i.e., with that of 30 valuable digits.

### 4.1.3. Evolution for $t \geq t_\Lambda$

For times $t \geq t_\Lambda$, the well-known classical evolution takes place. The time dependences of the cosmological variables $H(t)$, $\rho^{(f)}(t)$, and $a(t)$ are given in Equations (111), (112), and (113), respectively, rederived now in Scheme B. The constant of integration $C_t$ can be settled by requiring the continuity of $H(t)$ at $t = t_\Lambda$. With the accuracy of 30 valuable digits, one obtains $C_t \approx \frac{2}{3}t_G$. The explicit evaluations with the help of Formula (65) show again that the cosmological quantities considered here are continuous at $t = t_\Lambda$ at least with the accuracy of 60 valuable digits. It was checked analytically that $\rho^{(f)}(t)$ and $a(t)$ are also continuous with the same accuracy at $t = t_\Lambda$ for that choice of $C_t$. The constant of integration $C_t$ describes that the Big Bang singularity should be dated with $\frac{2}{3}t_G$ earlier due to quantum effects than it was extrapolated from the classical cosmology.

### 4.2. Scheme B

In this section, we show that the solution of the system of Equations (21)–(27) provides exactly the same evolution in the case of the particular model discussed by us as the one we obtained in the framework of Scheme A. For $0 < t \leq t_\Lambda$, we have five equations for three yet-unknown functions: $\Omega^{(f)}(N)$, $y(N)$, $k(N)$. In order to test that our interpretation of $\rho^{(\Lambda)}$ as a field-independent potential energy density of a homogeneous scalar field does not lead to any contradiction, we checked that the constraints (23), (24), and either Equation (21) or Equation (27) represent systems of three independent equations. Namely, both Equations (22) and (27) can be derived from the system of Equations (21), (23) and (24). In our particular model, the functions $y(k)$ and $\Omega^{(f)}(k)$ are directly determined from the constraints (23) and (24):

$$y^2(k) = -\frac{\eta_{\mathrm{RG}}(k)}{\nu_{\mathrm{RG}}(k)}, \quad \Omega^{(f)}(k) = 1 + \frac{\eta_{\mathrm{RG}}(k)}{\nu_{\mathrm{RG}}(k)}, \tag{82}$$

and the evolution $k(N)$ of the RG scale is governed by Equation (27).

### 4.2.1. Time Interval $0 < t \leq t_G$

Making use of the scale dependences given in Equations (2) and (3), we find that the RG parameters are the scale-independent constants $\eta_{\mathrm{RG}} = -2$, $\nu_{\mathrm{RG}} = 4$, and therefore, Equation (28) yields $\alpha_{\mathrm{RG}} = 1$, and the solution of the constraints (23) and (24) are the constants $y^2(= \Omega^{(\Lambda)}) = \Omega^{(f)} = \frac{1}{2}$. Thus, the solution in the time interval $0 < t \leq t_G$ represents a cosmological fixed point, when the matter content is a mixture of the barotropic fluid and the dark energy with equal energy densities. The dynamical Equation (27) for the RG scale reduces to

$$\frac{d\ln k}{dN} \equiv \frac{d\ln k}{d\ln a} = \frac{3\gamma}{2}\Omega^{(f)} = \frac{3\gamma}{4}, \tag{83}$$

having the solution:

$$k = \mathcal{K}_1 a^{-3\gamma/4} \tag{84}$$

with the constant of integration $\mathcal{K}_1$. Then, one finds from Equation (29) that $\frac{d\ln H}{dN} = \frac{d\ln k}{dN}$, so that

$$H = C_H k = C_H \mathcal{K}_1 a^{-3\gamma/4} \tag{85}$$

with the constant of integration $C_H$. Equation (85) with $H = \dot{a}/a$ represents a first-order ODE for the scale factor $a(t)$ having the solution:

$$a(t) = \left(\frac{3\gamma C_H \mathcal{K}_1}{4} t\right)^{\frac{4}{3\gamma}}, \qquad (86)$$

where the point of singularity has been shifted to $t = 0$ by the appropriate choice of the constant of integration. Inserting the time dependence of the scale factor (86) back into Equations (84) and (85), one obtains the explicit time dependence of the scale factor and that of the Hubble parameter:

$$k(t) = \frac{4}{3\gamma C_H} \frac{1}{t}, \qquad (87)$$

$$H(t) = \frac{4}{3\gamma} \frac{1}{t}. \qquad (88)$$

Then, the knowledge of the $k$-to-$t$ conversion rule enables one to make explicit the time dependence of the quantities $G(t)$, $\Lambda(t)$, $\rho_c(t) = \frac{3H^2}{8\pi G(t)}$, and $\rho^{(f)}(t) = \rho^{(\Lambda)}(t) = \frac{1}{2}\rho_c(t)$. The comparison of $\rho^{(f)}(t)$ and $\rho^{(\Lambda)}(t)$ evaluated in that manner with the expression (30) and the definition $\rho^{(\Lambda)} = \frac{\Lambda}{8\pi G}$, respectively, enables one to settle the yet-undetermined constants of integration:

$$\mathcal{K}_1 = \left(\frac{16\pi g_*}{\lambda_*} \mathcal{M}\right)^{1/4}, \quad C_H = \sqrt{\frac{2\lambda_*}{3}}. \qquad (89)$$

In this manner, we recapitulate all the results given in Equations (34)–(40) obtained in the framework of Scheme A. The additional point is that we could now identify this early evolution of the discussed model Universe with a cosmological fixed point with evolving RG scale.

4.2.2. Time Interval $t_G \leq t \leq t_\Lambda$

Making use of the RG scale dependences of the gravitational couplings given in Equations (2) and (3), one establishes that the RG parameters are scale-dependent in the interval $t_G \leq t \leq t_\Lambda$:

$$\eta_{\mathrm{RG}}(k) = \frac{-2k^2}{-k^2 + \frac{1}{2}E}, \qquad (90)$$

$$\nu_{\mathrm{RG}}(k) = \frac{2k^2(-k^4 + Ek^2 + F)}{(k^4 + F)(-k^2 + \frac{1}{2}E)}, \qquad (91)$$

$$\alpha_{\mathrm{RG}}(k) = 1 + \frac{\eta_{\mathrm{RG}}(k)}{\nu_{\mathrm{RG}}(k)} = \Omega^{(f)}(k). \qquad (92)$$

Then, Equation (27) yields the first-order ODE:

$$\frac{d\ln k}{dN} = \frac{3\gamma}{2} \qquad (93)$$

for the evolution of the RG scale, having the solution:

$$k = \mathcal{K}_2 e^{\frac{3\gamma}{2}N} = \mathcal{K}_2 a^{-\frac{3}{2}\gamma} \qquad (94)$$

with the constant of integration $\mathcal{K}_2$. Inserting the solution (94) into Equation (29), one finds the first-order ODE:

$$\frac{d\ln H}{dk} = \frac{\alpha_{\mathrm{RG}}(k)}{k} = 2k\frac{-k^2 + \frac{1}{2}E}{-k^4 + Ek^2 + F}, \tag{95}$$

having the solution:

$$H(k) = D_H\sqrt{-k^4 + Ek^2 + F} \tag{96}$$

with the constant of integration $D_H$. With the help of the $k$ dependences of the Hubble parameter (96) and the RG parameters (90), one is enabled to determine the $k$ dependences of the other important cosmological quantities:

$$\rho_c(k) = \frac{3H^2(k)}{8\pi G(k)} = \frac{3D_H^2}{8\pi b}\frac{-k^4 + Ek^2 + F}{-k^2 + \frac{1}{2}E}, \tag{97}$$

$$\rho^{(f)}(k) = \rho_c(k)\Omega^{(f)}(k) = \frac{6D_H^2}{8\pi b}k^2 = \frac{6D_H^2}{8\pi b}\mathcal{K}_2^2 a^{-3\gamma}, \tag{98}$$

$$\Omega^{(\Lambda)}(k) = y^2(k) = \frac{k^4 + F}{-k^4 + Ek^2 + F}, \tag{99}$$

$$\rho^{(\Lambda)}(k) = \rho_c(k)\Omega^{(\Lambda)} = \frac{3D_H^2}{8\pi b}\frac{k^4 + F}{-k^2 + \frac{1}{2}E}. \tag{100}$$

The comparison of Equations (98) and (100) with Equations (30) and (1), respectively, yields the expressions of the constants of integration:

$$D_H = \sqrt{\frac{\lambda_*}{9g_* m_{\mathrm{Pl}}^2}}, \quad \mathcal{K}_2 = \sqrt{\frac{8\pi}{\lambda_* m_{\mathrm{Pl}}^2}\mathcal{M}}. \tag{101}$$

Now, we can compare our results (66) and (96) for $H(k)$ obtained in the frameworks of Scheme A and Scheme B, respectively. Taking into account the definition (48) of $D(k)$ and the explicit expressions (49)–(51) of the constants $b_0$, $b_1$, and $b_2$, on the one hand, and the explicit expression of the constant $D_H$ given in (101) and the relations $E = 3b_1/c$, $F = 3b_0/c$, and $3b_2/c = 1$, on the other hand, one concludes that both schemes provide the same result for $H(k)$. Moreover, taking the time derivative of both sides of Equation (94), we just recover the first-order ODE (68) found in Scheme A. We have shown that the constraint (24) of Scheme B is identical to the reduced consistency condition (14) of Scheme A in the case of the particular model discussed by us. Then, the dynamical problem of evolution to be solved turns out to be identical in both schemes. Thus, we obtain the same time dependences of all cosmological quantities in both schemes in the time interval $t_G \leq t \leq t_\Lambda$.

4.2.3. Evolution for $t \geq t_\Lambda$

For $t \geq t_\Lambda$, the RG scale is frozen at $k = k_\Lambda$, and the evolution becomes classical, described by two independent equations: for example, the classical versions of Equation (21):

$$\frac{d\Omega^{(f)}}{dN} = 3\gamma\Omega^{(f)}(1 - \Omega^{(f)}), \tag{102}$$

and the Friedmann constraint (23). Furthermore, the equation holds:

$$\frac{d\ln H}{dN} = \frac{3}{2}\gamma\Omega^{(f)}, \tag{103}$$

which represents Equation (11) in [1] applied to our particular model Universe.

As a first step, one can determine the evolution of $\Omega^{(f)}$ vs. the parameter $N$ and express all cosmological quantities in terms of $\Omega^{(f)}$. The first-order ODE (102) can be solved in a straightforward manner by making use of the integral Formula 2.103.4 in [35]:

$$\Omega^{(f)} = (1 + C_\Omega e^{-3\gamma N})^{-1} = (1 + C_\Omega a^{3\gamma})^{-1} \tag{104}$$

with the constant of integration $C_\Omega$. Then, one obtains trivially from the Friedmann constraint that

$$y^2 = \Omega^{(\Lambda)} = 1 - \Omega^{(f)}. \tag{105}$$

Inserting the function $\Omega^{(f)} = \Omega^{(f)}(N)$ given in Equation (104) into the right-hand side of Equation (103), one obtains a first-order ODE for the $N$ dependence of the scale factor, having the solution:

$$H = \frac{C_H C_\Omega^{1/2}}{\sqrt{1 - \Omega^{(f)}}}, \tag{106}$$

reexpressed in terms of $\Omega^{(f)}(N)$, where $C_H$ is the constant of integration. Then, one finds by definition that

$$\rho_c = \frac{\rho^{(\Lambda)}}{y^2} = \frac{\Lambda_0}{8\pi G_0(1 - \Omega^{(f)})}, \tag{107}$$

$$\rho^{(f)} = \rho_c \Omega^{(f)} = \frac{\Lambda_0}{8\pi G_0} \frac{\Omega^{(f)}}{1 - \Omega^{(f)}}, \tag{108}$$

and from the second equality of Equation (104) that

$$a = \left( \frac{1 - \Omega^{(f)}}{C_\Omega \Omega^{(f)}} \right)^{\frac{1}{3\gamma}}. \tag{109}$$

Next, we determine the inverse function $t = t(\Omega^{(f)})$ by making use of the relation $dt = -dN/H$:

$$\begin{aligned}
t &= -\int \frac{dN}{H} = -\frac{1}{3\gamma C_H C_\Omega^{1/2}} \int \frac{d\Omega^{(f)}}{\Omega^{(f)} \sqrt{1 - \Omega^{(f)}}} \\
&= -\frac{1}{3\gamma C_H C_\Omega^{1/2}} \ln \frac{1 - \sqrt{1 - \Omega^{(f)}}}{1 + \sqrt{1 - \Omega^{(f)}}} + C_t
\end{aligned} \tag{110}$$

with the constant of integration $C_t$. (Here, we made use of the integral Formula 2.224.5 in [35].) Now, one has to settle the constants of integration. With the help of the relations (108) and (109), one expresses $\rho^{(f)}$ in terms of the scale factor $a$, and a comparison with the expression (30) yields then $C_\Omega = \frac{\Lambda_0}{8\pi G_0 \mathcal{M}}$. The constant $C_H$ is easily settled by requiring the equality of $\rho_c = \frac{3H^2}{8\pi G_0} = \frac{C_H^2 C_\Omega}{1 - \Omega^{(f)}}$ with the expression given in Equation (107). This implies $C_H^2 C_\Omega = \Lambda_0/3$ and $C_H = \sqrt{\frac{8\pi G_0 \Lambda_0 \mathcal{M}}{3}}$. Finally, we invert the relation (110) and obtain the well-known time dependences obtained generally by solving the problem of evolution in terms of the traditional, dimensionful cosmological variables:

$$H(t) = \sqrt{\frac{\Lambda_0}{3}} \left[ \tanh\left( \frac{3\gamma}{2} \sqrt{\frac{\Lambda_0}{3}}(t - C_t) \right) \right]^{-1}, \tag{111}$$

$$\rho^{(f)}(t) = \frac{\Lambda_0}{8\pi G_0} \left[ \sinh\left( \frac{3\gamma}{2} \sqrt{\frac{\Lambda_0}{3}}(t - C_t) \right) \right]^{-2}, \tag{112}$$

$$a(t) = C_\Omega^{-\frac{1}{3\gamma}} \left[ \sinh\left( \frac{3\gamma}{2} \sqrt{\frac{\Lambda_0}{3}}(t - C_t) \right) \right]^{\frac{2}{3\gamma}}. \tag{113}$$

The determination of the constant of integration $C_t$ is discussed at the end of Section 4.1.3.

*4.3. Typical Time Scales*

In Sections 4.1 and 4.2, it was shown that both schemes provide the same evolution of the discussed particular model Universe. This evolution is characterized by various time scales. The evolution following just the the Big Bang singularity is governed by the RG evolution of the gravitational couplings determined by the Reuter fixed point and ends up at $t_G = \xi^{UV}/k_G$ (see Equation (43)) corresponding to the dynamical RG scale $k_G$. The inverse proportionality $t = \xi^{(UV)}/k$ is valid in the whole time interval $0 < t \leq t_G$ with the coefficient $\xi^{(UV)} = \sqrt{\frac{8}{3\lambda_*}}\frac{1}{\gamma t}$ (see Equation (34)), in agreement with previous findings [30,31]. There exists another dynamical RG scale, $k_\Lambda$, such that, for $k \leq k_\Lambda$, the gravitational couplings keep their constant values. The corresponding time scale $t_\Lambda$ is given by Equation (64). Since $t_G/t_\Lambda \sim \mathcal{O}(10^{-30})$, one finds that $t_\Lambda \approx \xi^{(IR)}/k_\Lambda$ with an accuracy of 30 valuable digits, where $\xi^{(IR)} = \frac{2}{3\gamma\sqrt{\lambda_*}}$. Our finding for $\xi^{(IR)}$ is rather similar to that found in [29] in the perturbative regime, $\xi^{(pert)} = \sqrt{\frac{2\omega}{3\nu}}\frac{1}{\gamma}$, where $\omega$ and $\nu$ are RG scheme-dependent constants in the perturbative expansion of the gravitational couplings close to the Gaussian fixed point. The intermediate era of the evolution is governed by the crossover RG scaling properties of the gravitational couplings showed up between the UV and deep IR scaling regions. The $k$-to-$t$ conversion rule is rather complicated in that era, being not at all a simple inverse proportionality, but can be estimated with a good accuracy with the formula given in Equation (65).

In addition to the time scales $t_G$ and $t_\Lambda$, there are two other important time scales: (i) $t_d$, when the accelerating expansion turns into a decelerating one, and (ii) $t_a$, when the decelerating expansion turns again into an accelerating one. The time scale $t_a \sim \mathcal{O}(\Lambda_0^{-1/2})$ is a feature of the classical cosmology occurring due to the dominance of the nonvanishing cosmological constant $\Lambda_0$ in the late future. The scale $t_d$ is, however, affected by quantum effects. The deceleration parameter:

$$q = -\frac{a\ddot{a}}{\dot{a}^2} = -1 - \frac{\dot{H}}{H^2} \tag{114}$$

can easily be calculated for the Planck and the crossover eras. One finds

$$q = -1 + \frac{3\gamma}{4} = \begin{cases} 0 & \text{for} \quad w = 1/3 \\ -\frac{1}{4} & \text{for} \quad w = 0 \end{cases}, \tag{115}$$

for $0 < t \leq t_G$ and

$$q = -1 + \frac{3\gamma}{2} - \frac{2}{27 g_* \lambda_* \gamma m_{Pl}^2 \bar{t}^2} \tag{116}$$

for $t_G \leq t \leq t_\Lambda$. The deceleration parameter is continuous at $t = t_G$ and is a strictly monotonically increasing function of $t$ in the crossover era, taking the positive value:

$$
\begin{aligned}
q(t_\Lambda) &= -1 + \frac{3\gamma}{2} - \frac{\gamma k_\Lambda^2}{2 k_G^2} \\
&= \left\{
\begin{array}{ll}
1 - \frac{2}{3}\frac{k_\Lambda^2}{k_G^2} & \text{for} \quad w = 1/3 \\
\frac{1}{2}\left(1 - \frac{k_\Lambda^2}{k_G^2}\right) & \text{for} \quad w = 0
\end{array}
\right\}
\end{aligned}
\tag{117}
$$

when the crossover era is ended. One can see that the value of the deceleration parameter is model-dependent: for radiation ($w = 1/3$), it is vanishing at $t = t_G$, so that in that case, it holds $t_d = t_G$, but for dust ($w = 0$), $q$ changes sign at $t_d > t_G$. The condition $q(t_d) = 0$ yields that

$$
\begin{aligned}
t_d &= \frac{2}{3}t_G + \sqrt{\frac{\gamma}{12}}t_G\left(-1 + \frac{3\gamma}{2}\right)^{-1} \\
&= \left\{
\begin{array}{ll}
t_G & \text{for} \quad w = 1/3 \\
\frac{2+\sqrt{3}}{3}t_G > t_G & \text{for} \quad w = 0
\end{array}
\right\}.
\end{aligned}
\tag{118}
$$

## 5. Summary

In this paper, we proposed simple, analytic formulas to describe the main features of the RG scale dependences of the gravitational couplings in the framework of four-dimensional, two-parameter EH gravity. These analytic formulas interpolate in a continuous manner between the Reuter-fixed-point-governed UV scaling regime and the low-energy IR regime, where the couplings take their observed classical values. The interpolation formulas were constructed to ensure the continuity of the energy density of matter at the border of the UV and crossover scaling regions in the framework of asymptotically safe cosmology. We applied the proposed interpolation formulas to an analytically solvable, homogeneous, and isotropic, spatially flat model Universe, which contains a classical barotropic fluid and the $\Lambda$-component identified here with dark energy. The local energy conservation of the barotropic fluid has been required separately. The problem of evolution was solved in two schemes. In Scheme A, we followed the method used in [31], i.e., we worked in terms of the traditional, dimensionful cosmological variables and determined the conversion rule $k(t)$ between the RG scale $k$ and the cosmological time $t$ by means of the reduced consistency condition (14) of the quantum-improved Friedmann equations. In Scheme B, we followed the method used in [1], i.e., we worked in terms of dimensionless cosmological variables and determined the conversion rule $k(t)$ from the Friedmann-constraint-induced constraint on the RG parameters (24). We showed that the constraints (14) and (24) are equivalent in the case of the particular model. Therefore, both schemes provided just the same result for the evolution of the system.

In the framework of the particular model, explicit formulas were given for the characteristic time scales $t_G$ and $t_\Lambda$ corresponding to the dynamical RG scales $k_G$ and $k_\Lambda$, respectively, arising in the RG analysis of EH gravity. We recovered the well-known result of quantum-improved cosmology, that in the Planck era, just the inverse proportionality $t \propto 1/k$ holds. In the crossover era, a nontrivial relation was obtained for the conversion rule $k(t)$, but it happens that, at the end of the crossover era, the relation $t_\Lambda \propto 1/k_\Lambda$ holds with an extremely high accuracy. The characteristic time scale $t_d$ was also discussed, when the quantum effects driving the accelerating expansion of the early Universe go into a decelerating one. It was shown that $t_d \geq t_G$ holds and the value of $t_d$ depends on the matter content of the model. The determination of the solution in Scheme B enabled us to identify the cosmological fixed points, in the sense of the classification given in [1], from which the evolution starts and at which it ends up. As a by-product, it was found that the quantum effects result in the change of the entropy of the dark energy.

**Author Contributions:** Conceptualization, S.N. and K.S.; writing—review and editing, S.N. and K.S. All authors have read and agreed to the published version of the manuscript.

**Funding:** This research received no external funding.

**Data Availability Statement:** Not applicable.

**Conflicts of Interest:** The authors declare no conflict of interest.

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
