# Peer review of "Interpolation Formulas for Asymptotically Safe Cosmology"

_universe, doi:10.3390/universe9040184_

Round 1

Reviewer 1 Report

The paper is well done and, in my view, it shoud be accepted in the present form.

Author Response

We thank the referee for the positive report.

Sincerely,

Sandor Nagy

Reviewer 2 Report

The paper provides analytic formulas to describe the main features of
the RG scale dependences of the gravitational couplings in the framework of 4-dimensional, 2parameter EH gravity. I would like to ask the authors to clarify the following point.

 Are the analytical formulas presented in the manuscript expected to change in presence of a fluid (that dominates the Universe) having arbitrary equation of state parameter (Omega={-1,1}) ?

I would be happy to recommend the paper for publication, provided the above mentioned point is answered.

Author Response

We thank the referee for his/her valuable comments. We modified the manuscript in order to clarify this question. We highlighted the modifications by bold face letters in the lines (128-136) and also in (149-151). A new reference is added in [36].

Sincerely,

Sandor Nagy

Reviewer 3 Report

The paper is devoted to the construction of simple interpolation formulas for the description of the renormalization group scale dependences of the gravitational couplings in the framework of the 2-parameters Einstein-Hilbert (EH) theory of gravity and applied to a simple, analytically solvable, spatially homogeneous and isotropic, spatially flat model universe. There are interesting results.

The paper can be accepted fo publication after minor revision.

The article abounds with a lot of formulas, for a better perception it would be useful to illustrate the key formulas graphically. However, this wish remains at the authors' decision.

The paragraph represented by lines 114-125 requires some further explanation. Thus the authors introduce the EOS p=w\ro with 0<w<1. It would be interesting to know what matter corresponds to 1/3<w<1? It is obvious that lambda with w=-1 is not included here. Also the equation of state with w=-1 cannot correspond to scalar fields for which w>-1 and time dependent. The case w=-1 corresponds to only for the cosmological constant \Lambda.

Author Response

We thank the referee for his/her valuable comments. We made some modifications on the manuscript in order to clarify this issue. We highlighted the modifications by bold face letters in the lines (128-136) and also in (149-151). A new reference is added in [36].

Sincerely,

Sandor Nagy